# Educating for Virtuous Intellectual Character and Valuing Truth

Duncan Pritchard 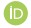

Department of Philosophy, University of California, Irvine, CA 92617, USA; dhpritch@uci.edu

**Abstract:** This paper explores the thesis that the overarching goal of education is to cultivate virtuous intellectual character. It is shown how finally valuing the truth is central to this theory on account of how such valuing is pivotal to intellectual virtues. This feature of the proposal might be thought to be problematic for a number of reasons. For example, it could be argued that truth is not valuable, that insisting on valuing the truth in educational contexts could be politically dubious, or that there is something unduly prescriptive about an educational methodology that has this component. It is argued, however, that many of these grounds for concern are not sound on closer inspection. Properly understood, educating for virtuous intellectual character, even once the truth-valuing aspect of this thesis is made explicit, should not be a contentious proposal.

**Keywords:** education; epistemology; intellectual virtue; truth; axiology

## 1. Introduction

My concern is in the idea that an overarching goal of education is the cultivation of virtuous intellectual character. In particular, I am interested in the specific role that finally (i.e., non-instrumentally) valuing the truth plays in this proposal and how this aspect of the view might be thought to be problematic. This educational thesis goes right back to antiquity; indeed, it can be found in some form in several classical intellectual traditions, but it is also a proposal that has gained common currency in recent theoretical work in education. My goal here is to set out what this stance involves, especially with respect to its axiological commitment to the truth, and to consider some of the main challenges that it faces, which are associated with this axiological commitment.

## 2. Educating for Virtuous Intellectual Character

Educating for virtuous intellectual character means conceiving of one's educational practices as being centrally concerned with the cultivation of an integrated set of admirable intellectual character traits, known as intellectual virtues. Examples of intellectual virtues include high-level cognitive traits such as intellectual humility, intellectual courage, conscientiousness, and intellectual tenacity. This general idea has an illustrious history, with versions of it found not only in ancient Greek and Roman thought, but also in other classical traditions as well, such as Confucianism[1]. The impetus for our current discussion, however, is the contemporary rediscovery of this idea. There is now a wealth of theoretical work on this topic alongside applied projects that aim to put these theoretical ideas into action in the classroom[2].

Notice that the focus here is specifically on the intellectual virtues. This is important because there is a virtue-theoretic educational program with a much broader focus that is concerned with the cultivation of the virtues in general, whether intellectual, moral, or otherwise (such as practical or civic)[3]. Our concern, however, will only be on the role of intellectual virtues in the educational enterprise.

What does it mean to claim that our educational practices should be *centrally* concerned with the cultivation of virtuous intellectual character? I take this to mean that an overarching goal of education is to achieve this aim. Obviously, education serves lots of

purposes, including practical, social, moral, political, and so on. But one core purpose of education is clearly epistemic in that it is designed to generate epistemic goods and skills such as knowledge, understanding, good reasoning, and so forth. Indeed, the epistemic purpose of education is clearly fundamental to the educational enterprise, as a putatively educational activity that did not serve any epistemic purpose—which, for example, only led to misunderstanding, ignorance, and poor reasoning—could not be a genuine educational activity at all. Relatedly, the other purposes of education would be undermined if the epistemic component was lacking. For instance, the broadly civic role of education in a democratic society of producing informed citizens would be undermined if the education system did not in fact inform people.

We can thus say that an overarching goal of the educational enterprise is epistemic in nature. But what is the nature of this overarching epistemic goal? We have noted that education seems to be directed towards generating a range of epistemic goods, but some of these epistemic goods appear to be more fundamental from an educational point of view than others; this is where virtuous intellectual character comes in. For it seems that while education, properly conducted, is obviously geared towards producing familiar epistemic goods and skills such as knowledge and good reasoning, how these epistemic goods and skills are generated is crucially important. We do not think an appropriate educational strategy would involve coercion or threats, for example, even if that would be the most effective way of ensuring that students master the knowledge and skills assigned to them. Rather, we want to enable students to be autonomous learners who acquire the high-level intellectual skills that ensure that they can acquire the knowledge and cognitive abilities that they need, and appropriately employ the knowledge and cognitive abilities that they have. As we will see, these high-level intellectual skills are the intellectual virtues, the components of virtuous intellectual character. This is thus the sense in which educating for virtuous intellectual character is central to the educational enterprise, as an overarching goal of this enterprise is epistemic in nature, and within that goal, the cultivation of virtuous intellectual character is of primary importance[4].

With this in mind, let us consider the nature of intellectual virtues. They have a number of important properties. One key property that will be particularly significant for our purposes is that intellectual virtues incorporate a distinctive motivational state, that of desiring and valuing the truth (what I have termed elsewhere as 'veritic desire')[5]. This means caring about accuracy, about getting things right, and thus, aiming to avoid falsity and error. This desire for truth also entails seeking knowledge and understanding of the subject matters that one is concerned with and, in the process, dispelling ignorance about those subject matters. This is because in attaining knowledge and understanding of a subject matter, one is thereby acquiring an integrated grasp of a body of truths and is gaining a firmer grip on the truth in the process. This is why the intellectually virtuous subject is engaged in open-ended inquiry, as their desire for the truth means that in answering any one line of questioning, they are invariably led onto a new line of inquiry that extends out from the last, with each further line of inquiry deepening their understanding and hence their grasp of the truth[6].

It is important that this valuing of the truth is not simply instrumental in nature. To be intellectually virtuous is to recognize that the truth, while undoubtedly useful, can also be worth seeking for its own sake (and thus has final value). The intellectually virtuous person is someone who cares about getting things right, even when there is no practical benefit. Relatedly, they seek to avoid falsehood and error, even when this might not matter from a practical point of view. In their desire for the truth, they are akin to a craftsperson who delights in producing a beautiful piece of furniture or a gardener who takes pleasure in being at peace with nature. In this way, the intellectually virtuous person is not someone who takes a cavalier attitude towards accuracy, nor are they the kind of person who views getting to the truth and avoiding falsity as merely strategic concerns; instead, truth is an important good to be prized alongside other finally valuable goods.

This veritic desire is crucial to an intellectual virtue; a cognitive trait that lacks this motivational state could not be an intellectual virtue. For example, acting in all respects as if one is intellectually humble, but doing so for a motivation that has nothing to do with one's concern for the truth (but motivated instead, say, by a desire to be admired by one's peers) would not be manifesting the intellectual virtue of intellectual humility. Similarly, it is not enough for a manifestation of an intellectual virtue such that one cares about the truth, but only in an instrumental fashion. If one's apparent intellectual humility is only driven by one's concern for the truth to the extent that caring about the truth in this way will make one admired by one's peers, then one is not really being intellectually humble at all, but merely seeming to be so.

That the intellectual virtues require this motivational state sets them apart from many of our other cognitive traits. One does not have to care about the truth at all in order to skillfully perceive one's environment, for example (a fortiori), one does not have to care about it in a non-instrumental fashion). More generally, another difference between intellectual virtues and our cognitive traits is the distinctive way that the former items are acquired. We are not born with intellectual virtues, nor are they the kind of cognitive trait that one gains naturally as part of one's biological development (for example, as one's cognitive faculties mature). Instead, we must consciously develop them. Moreover, we have to keep cultivating them thereafter; they are not the kind of cognitive skill that, once acquired, is easily retained.

Cultivating one's intellectual virtues often requires one to emulate exemplars who are already intellectually virtuous. The goal is to make one's manifestation of intellectual virtue second nature, such that through a process of habituation, one is able to unreflectively be intellectually virtuous in one's day-to-day life. Accomplishing this requires one to develop the good judgement to navigate between the two intellectual vices that correspond to each intellectual virtue, an intellectual vice of excess and an intellectual vice of deficiency. For example, being intellectually humble involves avoiding the intellectual vice of deficiency, whereby one is lacking in intellectual humility and hence is arrogant and dogmatic. But it also involves avoiding the intellectual vice of excess, whereby one is completely lacking in intellectual self-confidence. The target intellectual virtue lies on the 'golden mean' between these two polar intellectual vices[7].

A final property of the intellectual virtues that is relevant for our purposes is their special value. For while the intellectual virtues, like most cognitive skills, are practically useful (and hence instrumentally valuable), they are also in addition held to be of final, or non-instrumental, value. This is because they are thought to have *eudaimonic* value as they promote the good life of human flourishing. If so, then being intellectually virtuous is one of the few things that is good for its own sake, independently of its practical utility (though being intellectually virtuous is certainly held to be practically useful as well).

That the intellectual virtues are not just instrumentally valuable, like most other cognitive abilities, but also finally valuable would in part explain why one might want to make their cultivation central to the educational enterprise. For do we not envisage education when taught well as not merely training people up with useful skills and knowledge, but also helping them to grow and thrive as individuals? If so, then that goal would be best served by incorporating *eudaimonic* goods like intellectual virtues into one's pedagogical practices.

But even setting aside the putative final value of intellectual virtues, there is also the special kind of instrumental value that they possess; as we might put the point in the contemporary language of professional development, the intellectual virtues are the most transferrable kind of cognitive skill available. While most cognitive skills are tied to particular kinds of intellectual activity, the intellectual virtues are highly general cognitive traits that are applicable to any specific intellectual task that one is preoccupied with. No matter what one's intellectual concerns might be, it will be useful to engage in them in an intellectually virtuous manner.

This feature of the intellectual virtues relates to another distinctive property of them, which is the managerial role that they play in one's cognitive architecture. Having useful cognitive skills and knowledge is obviously beneficial, but the true benefit of any cognitive trait or knowledge lies in having the good judgement to use it wisely. For example, the advantages of having excellent analytical abilities will be severely undermined if one puts them to pointless ends or, worse, to counterproductive ones. (On the latter front, for example, think of how a pedantic kind of intelligence can alienate people from those around them, including potential collaborators in cognitive projects). This is where the intellectual virtues—and virtuous intellectual character more generally (i.e., the integrated set of intellectual virtues)—becomes especially valuable, as it guides the use of one's cognitive abilities and knowledge. The intellectually virtuous person does not just have cognitive skills and useful knowledge, but is able to put these skills and knowledge to good purposes that will benefit the agent's personal growth[8].

One can thus see the attraction of educating for virtuous intellectual character because it provides students with not just an extra kind of cognitive skill, but with a stratum of higher-level cognitive abilities that are crucially important to one's personal and intellectual development. Indeed, an educational practice that did not incorporate a concern for the cultivation of intellectual character would be in danger of leaving students lacking in vitally important cognitive expertise.

### 3. The Challenge from Critical Thinking

There are a number of potential challenges to the idea of educating for virtuous intellectual character. I will be focusing on a family of such objections that I suggest ultimately arises out of the feature of intellectual virtues that they involve a final valuing of the truth[9].

Let us begin with one prominent challenge to educating for virtuous intellectual character that is found in the work of Harvey Siegel [40–44]. Siegel is known for advocating a very different kind of educational focus on intellectual character, one that targets a kind of critical rationality that has reason and critical thinking at its heart rather than intellectual virtues. There is, of course, a great deal of overlap between critical thinking skills and intellectual virtues. Nonetheless, educating for virtuous intellectual character is the more demanding educational model, as Siegel himself notes. In particular, while the intellectually virtuous individual will have good critical thinking skills, the converse does not follow, in that while those with good critical thinking skills will often be intellectually virtuous, this need not always be so.

Siegel has defended his proposal in light of this alternative account by arguing that there is something dubiously prescriptive about educating for virtuous intellectual character, in a way that does not afflict his own stance. Here, he is setting out his position, with the relevant contrast in this passage being the proposal to educate for intellectual virtue that he is critiquing:

> "If we are serious about treating students with respect, what they become and what dispositions and virtues they value, possess, and manifest is importantly *up to them*. While we strive to foster [*critical thinking*] abilities and dispositions in our students, we also (if we are doing it right) invite them to evaluate for themselves the worthiness of these things and submit our arguments for that worthiness to their independent scrutiny and judgment." ([43], p. 108)

Siegel's suggestion is that the thinner conception of intellectual character in terms of critical thinking capacities that he offers is not wedded to any particular way in which a student should think but is merely focused on helping students to think. In contrast, he claims that educating for virtuous intellectual character is prescriptive in just this way, as it incorporates a conception of how the students ought to be and what values they should have. In this fashion, Siegel claims that his approach is better able to respect students' autonomy in that

it leaves it entirely up to them what conclusions they might reach in their reasoning or what values they might end up endorsing.

This criticism of educating for virtuous intellectual character does not stand up to closer scrutiny. Let us start with the idea that such an approach undermines students' autonomy by directing students towards particular ways of being rather than leaving this a matter for their own considered judgement. To begin with, we should remember that educating for virtuous intellectual character does not bring with it any particular factual commitments at all; it concerns ways of thinking, but not what one thinks. Siegel does not suggest otherwise, but it is worthwhile to remind ourselves of this point regardless, as it closes off one way in which an educational practice might be prescriptive in its outcomes in the kind of problematic fashion that Siegel has in mind.

What about the suggestion that educating for virtuous intellectual character is prescriptive in the sense that it aligns students with a particular set of values? The problem with this suggestion is that educating for intellectual virtue only incorporates one value, the value of truth, and thus a respect for such things as accuracy, reasons, evidence, and argument[10]. It is hard to see, however, how such a value could be controversial, especially in an educational context. After all, if one strips an educational practice of a concern for truth in this sense, then it is unclear as to what remains as an educational practice at all. What would it mean to be teaching something without having any concern for getting things right? Relatedly, if the idea of truth as a value is problematic, then it will be just as much an issue for Siegel's own critical thinking approach as it would be for educating for intellectual virtue. How could one possibly put reason and critical thinking at the heart of one's educational practice without also imbuing that practice with a valuing of truth and all that comes with it?

It is interesting that in the passage cited above, Siegel's focus is not on the desire for truth that is at the heart of the intellectual virtues, but rather on the specific virtues themselves. That is, he clearly thinks that educating in a way such that we are treating particular intellectual character traits as traits to be prized is counter to a student's autonomy, as it is prescribing that students should develop these character traits rather than leaving this as a matter for their own judgement. Notice, however, that Siegel is also prizing certain intellectual character traits on his approach, for how else are we to understand his proposal that we should educate for critical thinking skills? Moreover, it is interesting that in this passage, Siegel seems to be treating intellectual autonomy as being of paramount value, which presumably also has implications for the kinds of intellectual character traits that students should be acquiring on this proposal. If Siegel's concern here is valid, it thus has just as much application against his own proposal, given that it also has axiological commitments with implications for the intellectual traits that students are meant to acquire. In any case, a concern for good intellectual skills, whether critical thinking skills or intellectual virtues, flows naturally from the valuing of the truth. If we value the truth, then we value the kinds of intellectual skills that help us to get to the truth. Accordingly, insofar as we treat the latter as an inevitable feature of the educational enterprise, then it is no surprise that it will lead us to incorporate the teaching of these intellectual skills as well.

Still, a defender of Siegel's approach could insist that there can be ways of approaching this educational goal that are more theoretically loaded than others and that the benefit of his stance is that it carries much less baggage in this regard. For sure, students cannot develop their critical reasoning skills without caring about the truth at all, but they do not need to care about it in the way that the intellectually virtuous person does. In order to see this point, consider an artful lawyer. In order to be an expert in this field, it is obviously vital that one respects the truth; one needs to be able to keep track of the facts, marshal the arguments, follow them where they lead, and so on. Even so, one's concern for the truth could be entirely instrumental in nature, in that one cares for it only because it helps one win cases, gain promotions, earn the esteem of one's colleagues, and so on. One might not care about the truth in itself at all. And yet, as we noted earlier, this is not how the veritic desire that is built into the intellectual virtues manifests itself, as that does involve a valuing

of truth for its own sake. This is thus one core sense in which educating for the intellectual virtues brings with it axiological commitments that are absent from educating for critical thinking capacities, despite the obvious overlaps between these two theoretical approaches.

With this point in mind, Siegel may seem to be on much stronger ground, because why should we want to build into the educational enterprise that we should value truth in this more robust sense? Perhaps students will be led to that conclusion, but whether they are so led should be ultimately up to them, and we should not pre-judge the matter. Moreover, without that more robust value in play, then it similarly will also not be a foregone conclusion that students should develop the intellectual virtues specifically rather than just the more minimal critical thinking skills that Siegel advocates.

While Siegel is right that there is a substantive distinction at play here between educating for the intellectual virtues and educating for critical thinking capacities that we should be alert to, in highlighting this distinction, he also inadvertently demonstrates the need to opt for the former over the latter. In order to see this, consider what it would mean, in practice, to explicitly educate in such a way that truth is only to be valued in a purely instrumental fashion; for example, the truth is extolled as being an important good, but only because, and only insofar as, it serves your practical purposes. It is valuable because it helps you get better grades, for instance, and not because there is any inherent goodness in finding out things and thereby gaining understanding. Would it really be an attractive feature of an educational practice if it encouraged this kind of thinking about truth? Indeed, notice that on this proposal where truth lacks a practical value, it ceases to be worth having. So, if one can pass the exam without understanding the material, such as by parroting answers that make little sense to you, then why not?

The crux of the matter is that the kind of valuing of truth that we have in mind when we engage in good educational practices—when we cultivate intellectual character—does seem to be specifically of the more robust kind that is associated with the intellectual virtues and not the purely instrumental kind of valuing that Siegel has in mind. We want our students to take delight in gaining knowledge and understanding of the world around them and not merely seek knowledge because it serves some immediate practical purpose such as gaining oneself a better grade on an exam. Siegel is certainly correct, however, that once we do opt for the more robust kind of value of truth in our educational practices, then that will lead to strategies that favor educating for the intellectual virtues over educating for critical thinking capacities. If we train our students to be scholars who respect accuracy and aim to avoid falsehood in their intellectual activities, then we would expect these students to not only develop their critical thinking capacities but also end up cultivating their intellectual virtues.

Let me briefly summarize my position. On the one hand, I am agreeing that there is a fundamental value involved in educating for virtuous intellectual character, viz., the value of truth, where this is robustly understood such that it includes valuing the truth non-instrumentally. On the other hand, I am suggesting that this should not be a controversial value to incorporate into one's educational stance. In particular, I have claimed that this value is a necessary element of any plausible rendering of what we are aiming for in our educational practices, as any such rendering that lacks this value would be deficient in crucial respects[11].

## 4. Further Challenges Associated with Valuing the Truth

We have seen that Siegel's challenge against educating for virtuous intellectual character ultimately came down to the particular way in which the intellectually virtuous values the truth. We have argued, however, that this element of intellectual virtues is in fact essential to a plausible conception of the educational enterprise. Are there independent grounds to doubt that valuing the truth should play this role in our educational practices?

One way in which one might dispute the final value of truth is by arguing that there is no such thing as objective truth as we ordinarily suppose. Take an empirical issue such as whether Lee Harvey Oswald shot Kennedy or whether the planet Venus is larger than

the planet Mercury. We would naturally suppose that such issues are entirely settled by what is in fact the case; that is, either Lee Harvey Oswald shot Kennedy or he did not, but one of these options (the true one) is a historical fact, which only depends on what actually happened. Similarly, either the planet Venus is (right now) larger than the planet Mercury or it is not, and one of these options (the true one) is a cosmological fact, which only depends on the sizes of the relevant objects. In particular, it makes no difference to such truths whether we think that they are true (or not), whether we want them to be true (or not), whether we care whether they are true (or not), whether it would be advantageous to us if they were true (or not), and so on.

It might be held, in contrast, that truth is never objective in this way[12]. A relativist about truth might contend, for example, that it can be true for one set of people that Lee Harvey Oswald shot Kennedy and also true for another set of people that he did not. On this view, after all, truth can be relative to some further factor, such as what one's peer group happens to believe, and hence, opposing claims can both be true relative to different frames of reference. Relativism about truth is obviously a contentious thesis; indeed, it is almost certainly self-refuting[13]. But we can set that to one side for our current purposes and only focus on the challenge that it poses to educating for virtuous intellectual character. For the idea that truth is ever finally valuable clearly depends on it being the kind of thing that can be an objective property. No one would regard the truth as a finally valuable good if the relativist is right. (Indeed, it does not even seem to be instrumentally valuable on this proposal, as having a true belief is consistent with someone else believing the exact opposite claim also has a true belief. But how could it be that it is practically beneficial for both parties to have true beliefs if our respective true beliefs are contradictory?) Accordingly, if one rejects the objectivity of truth, then one will not be persuaded by the idea of educating for virtuous intellectual character, given that this involves treating truth as finally valuable.

What is interesting about this objection is that it is not really a challenge against educating for virtuous intellectual character specifically, but rather a challenge against the very idea of education. We noted earlier that education has a fundamental epistemic component in that part of what makes an educational strategy what it is—and not merely a form of indoctrination, say—is that it is concerned with propagating epistemic goods such as knowledge and understanding. But if there is no objective truth, then there is nothing to know or understand. Consider the putative contrast between education and indoctrination. If there is no objective truth, then why could a tyrant's indoctrination of their citizens not count as a genuine educational practice? The claims drilled into this populace would be true 'for them' after all. The crux of the matter is that treating truth as an objective matter is crucial to any plausible conception of education; hence, it ought to be uncontroversial that educating for virtuous intellectual character, on account of how it regards truth as finally valuable, is also committed to treating truth as an objective matter.

A more subtle way of objecting to educating for virtuous intellectual character might be to argue not that truth is not objective, but that we do not know which claims are actually true. The thought in play here is that there is a tension between, on the one hand, the idea that truth is finally valuable and we should virtuously seek it, and, on the other hand, the idea that we should educate for the intellectual virtues. In fact, the former is alleged to motivate radical skepticism. In caring about the truth and in virtuously seeking it, we should come to realize that we never know when we possess it. But educating for virtuous intellectual character is meant to be a way to structure the pedagogical transmission of knowledge and understanding, which obviously presupposes that we have knowledge and understanding. The two claims are thus in apparent conflict with one another. Indeed, it seems that the advocates of educating for virtuous intellectual character are not following their own advice, but are rather dogmatically (and thus vicefully) clinging to their belief that they have knowledge that they in fact lack.

Let us start by considering whether there is a case to be made for radical skepticism of this kind. While there are some compelling arguments for radical skepticism in the literature, they are not of much use to the proponent of this line, as they are employed

to motivate this thesis as a paradox rather than as a position (i.e., as a theoretical puzzle to be resolved rather than as a stance that one advocates)[14]. Our envisaged objection is clearly being offered in the latter vein, however, so these arguments will be unavailable here. The kind of arguments for skepticism that remain will not motivate radical skepticism, however; at most, they will motivate a localized version. Moreover, these are local skeptical arguments that should be very amenable to the proponent of a virtue-theoretic educational strategy.

For example, one might appeal to our fallibility as inquirers. That presents us with grounds to be doubtful of a range of claims that we might ordinarily be quite confident about, given that we are aware that our confidence is often misplaced; this proposal does not show that we should be skeptical of all our knowledge, however. Firstly, this line of argument proceeds by appealing to our knowledge of our fallibility, such as our knowledge of the specific ways in which we have previously gone astray in our judgments and the mechanisms of how this has occurred (cognitive bias, say). Accordingly, it can hardly be the case that a skeptical strategy of this kind is universal in scope. In any case, the appeal to fallibility is not plausible as a radical skeptical tactic anyway, because why should the mere fact that one's beliefs are fallible entail that they never amount to knowledge? To suppose that it does seems to appeal to an absurdly demanding infallibilist conception of knowledge (whereby one can only acquire knowledge through epistemic routes that could never lead to error), one so divorced from our ordinary, fallibilist, conception of knowledge so as to be little more than changing the subject[15].

With our fallibility-based skepticism understood in a purely localized fashion, however, it is not something that anyone who defends educating for virtuous intellectual character should be afraid of. This is because this is a form of skepticism that the intellectually virtuous subject will be naturally inclined towards. Consider, for example, one of the central intellectual virtues, that of intellectual humility. One key element of intellectual humility is recognizing one's own fallibility and thereby being open to the possibility of error[16]. It is thus built into the very idea of educating for virtuous intellectual character that one should be alert to ways in which one might be in error, and this will inevitably lead to a mitigated form of skepticism about the scope of one's knowledge as a result. Educating for the intellectual virtues is thus not in tension with localized skepticism of this kind.

Significantly, however, as we noted earlier, it is also central to an intellectual virtue that one should avoid *both* of the corresponding intellectual vices. When it comes to intellectual humility, this means not merely avoiding the intellectual vice of deficiency involved in dogmatism but also the intellectual vice of excess involved in doubting everything, even when one lacks an adequate rational basis for that doubt. When properly understood, then, it is not educating for virtuous intellectual character that incorporates a problematic attachment to the idea that we can know the truth. In contrast, those who attempt to advocate for the coherence of a widespread skepticism do seem to exhibit a problematically viceful intellectual character[17].

A different critical line towards educating for virtuous intellectual character in the same general vein is that there is something *politically* dubious about valorizing truth. Does this educational approach not overlook the way in which truth claims can manifest, often in quite hidden ways, important power relations? Accordingly, in saying that we should care about the truth, should we not first ask *whose truth* it is that we are caring about? The thought is thus that there is a certain political naivety built into this educational program, of a kind that could be quite problematic in pedagogical contexts.

This line of objection not only fails to lay a glove on our approach, but in fact highlights some of its strengths as an educational strategy. To begin with, the objection forgets that educating for virtuous intellectual character, while it leads to a valuing of the truth, is not committed to any particular set of truths. The point of the strategy is to help students learn to think for themselves rather than to tell them what to think; this is precisely one attractive feature of it. In particular, the intellectual virtues will provide students with the skills to critically assess the materials that they are given, including the various truth claims that

are presented to them. As we have noted, this kind of critical engagement has an abiding respect for truth at its heart, but this does not involve caring about particular truth claims so much as embracing a way of thinking that involves a love of the truth. The intellectually virtuous inquirer wants to get things right, which means understanding how things really are, properly capturing what different people's views amount to (i.e., not misrepresenting them), being alert to new evidence, including evidence that counts against one's current position, and so on. The point is that these are all intellectual skills which, if mastered, would provide one with the ability to differentiate between what is merely claimed as true and what is in fact true. In particular, it will provide students with the best means to accomplish this, even when the people presenting the truths are powerful. It follows that far from the claim of the project of educating for virtuous intellectual character being naïve about the political nature of truth, it is in fact the best educational strategy available for giving students the tools to engage with this problematic feature of the way appeals to truth can be employed.

I think this last kind of worry often goes hand-in-hand with a kind of ad hominem charge against educating for virtuous intellectual character. It is that this approach is usually associated with ancient Greek thought and thereby with an overarching Greco-Roman intellectual tradition that is thought to be the foundation of 'Western' civilization. The worry is that the history of this idea will ensure that it simply cannot be part of a decolonized curriculum, but will instead always end up buttressing existing political structures.

One should concede to this objection that educating for virtuous intellectual character could indeed be employed in a way that embeds it within a particular political structure; it is, after all, true of *any* educational strategy that it might be so embedded. For the charge to have teeth, however, it then needs to be the case that this approach is particularly well-suited to supporting the current political status quo; that, I suggest, is false. For one thing, educating for the intellectual virtues, at least if properly taught, would not sit well alongside any political orthodoxy. This is because those virtues are designed to give students the skills to critically engage with everything that is presented to them, no matter the authority of those doing the presenting. Accordingly, one should not expect students educated in this manner to owe fealty to any prevailing political wisdom.

Moreover, we should differentiate between educating for virtuous intellectual character and teaching students about the theory behind such a proposal, including its history. There is no essential reason why the former should be wedded to the latter. And even when it is so wedded, perhaps because one wishes to give more advanced students a deeper scholarly context in which this learning is taking place, there is no reason why one should discuss this context in a narrow and uncritical way. For example, there is a wealth of literature that critiques the way in which the narrative about 'Western' intellectual thought developed and the purposes to which it was devoted[18]; hence, if one wishes to teach the history of the intellectual virtues in ancient Greek thought alongside the intellectual virtues themselves, then it will be useful to also bring this critical scholarship into the mix as well[19].

Furthermore, insofar as one is concerned to provide students with a historical background to the intellectual virtues, then it would clearly be relevant to explain how very similar ideas arise in a range of intellectual traditions. To take one prominent example, a number of commentators have noted the striking overlaps between the Aristotelian conception of the intellectual virtues and Confucian ideas[20]. Accordingly, any attempt at educating for virtuous intellectual character that elects to explore the historical context of this proposal would be wise to explore how virtue-theoretic ideas arise in a range of intellectual traditions beyond the Greco-Roman tradition[21].

## 5. Concluding Remarks

We have seen that educating for virtuous intellectual character is a defensible educational program. In particular, while it is true that finally valuing the truth is at the heart of this proposal, on account of how it is central to the intellectual virtues, this axiological commitment is not contentious on closer inspection[22].

**Funding:** John Templeton Foundation ('Embedding the Development of Intellectual Character within a University Curriculum', #62330).

**Institutional Review Board Statement:** Not applicable.

**Informed Consent Statement:** Not applicable.

**Data Availability Statement:** Not applicable.

**Conflicts of Interest:** The author declares no conflict of interest.

## Notes

1    For an excellent contemporary discussion of the Aristotelian account of intellectual virtue and, more generally, its relationship to the virtues, see [1]. For some recent discussions of the connections between Confucian thought and the broadly Aristotelian conception of the intellectual virtues, see [2–6]. While not a classical precursor of contemporary accounts of educating for virtuous intellectual character, the work of the American pragmatist John Dewey is an important influence; see, especially, [7–9].

2    For some important contemporary theoretical work on educating for virtuous intellectual character, see [10,11] and the essays collected in [12]. See also [13–20]. For a useful recent survey of recent work on this topic, see [21]. I have previously explored the application of this theoretical program to two educational projects, in prison and in higher education—see, respectively, [22–24].

3    For some prominent recent discussions of general virtue-theoretic educational proposals of this kind, see [25–27].

4    For further discussion of the epistemic goals of education, see [13,15,17].

5    See [28].

6    I have explored this point about how desiring the truth entails desiring knowledge and understanding (i.e., as opposed to the latter being in some way distinct from the former) in a number of places. See, for example, [28–30].

7    For further discussion of the manner in which the intellectual virtues are acquired, including the role of exemplars in this respect, see [20,31].

8    For some important contemporary treatments of the intellectual virtues, see [1,32,33]; see also [30]. For a useful survey, see [34].

9    There are, of course, other objections to educating for virtuous intellectual character that attack the proposal from other directions. For example, there is the charge that this proposal is cognitively implausible as an educational model with concrete application. See, for instance, [35]. Relatedly, there is the complaint that there are inherent problems associated with measuring the development of intellectual virtue, which if true would obviously severely hamper any educational program geared towards this development. See [36,37]. There is also the more general scepticism about the existence of virtuous cognitive traits—whether moral, intellectual, or practical—that one finds amongst those defending situationism. See, for example, [38] for a specific rendering of the situationist challenge in the context of the intellectual virtues. For a response that specifically relates to the epistemology of education, see [39].

10   Arguably, educating for virtue simpliciter would likely bring in higher value claims than this, especially when it comes to educating for the moral virtues, but this is not the approach that is being advocated here.

11   For a related critical discussion of Siegel's critique of educating for virtuous intellectual character, see [45].

12   Of course, *some* truths are not true in this way; judgments of taste, for example, are not objective in this sense. If I judge that sourdough is poor tasting bread, there is no fact of the matter that settles this issue (though it is still a fact of the matter that I regard sourdough bread as poor tasting).

13   For a helpful recent overview of the status of truth relativism, see [46].

14   For an accessible overview of the radical sceptical debate, including the issue of whether radical scepticism is conceived of as a paradox or a position, see [47].

15   To use a famous example offered in [48], (ch. 2), it would be akin to defining a 'physician' as someone who can cure any illness within 24 h and then proceeding to argue for a scepticism about whether there are any 'physicians' in New York.

16   Indeed, on the leading account of this intellectual virtue, this is essentially what is involved in being intellectually humble. See [49]. My own view, for what it is worth, departs from this account in some important respects (albeit not in ways that are relevant to our current discussion). See, for example, [50].

17   For further discussion of the idea that being intellectually virtuous goes hand-in-hand with a kind of moderate localized scepticism, see [47], (ch. 4).

18   See, for example, [51] which offers a penetrating discussion of the dubious origin story that has been given to 'Western' philosophy, whereby philosophy begins with the ancient Greeks (dubbed retrospectively as 'Westerners') who somehow managed to completely invent philosophy thought ex nihilo and then develop it without any need for assistance from any other cultures (a feat that not even the ancient Greeks imagined they had pulled off). See also [52] which additionally explores some of the important precursors to ancient Greek thought in northern Africa.

<sup>19</sup> placeholder

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
