# Peer review of "Educating for Virtuous Intellectual Character and Valuing Truth"

_philosophies, doi:10.3390/philosophies8020029_

Round 1
Reviewer 1 Report
This is a well argued article that is ready for publication.
Author Response
I am grateful for the very positive comments on my paper. Since no specific changes were requested, I've not made any changes in response to this particular referee report.
Reviewer 2 Report
The paper provides an excellent argument for valuing truth in education and gives powerful counterarguments to alternative claims. I have no major comments. One thing is that the last objection may be a bit of a strawman. While few would argue that truth should not be valued in education because absolute truth is unachievable, one may also argue that truth often is out of reach. As a result the best we can usually aim for is justified belief or empirical accuracy or still some other mental state. Because this is usually all we can hope for, valuing truth could be too strong.
I encourage the author to engage with this revised objection. For the remainder, the paper is excellent.
Smaller comments in attached pdf.

Author Response
I am grateful for the referee's very positive comments on my paper. I've responded to all the specific comments made in the revised version of the paper. I haven't responded to the general comment made below, however, as I'm unsure what the referee has in mind. In particular, I don't know what they mean by 'absolute truth'. I don't employ this notion in the paper at all; my concern is only with an everyday sense of truth (getting things right, being accurate, and so on). The proposed 'weakening' of the truth goal by aiming for JB that the referee suggests also wouldn't work on my proposal, as that would also count as aiming for truth on my conception. In seeking justification one is seeking evidence that appropriately supports specific claims, after all, and that means being sensitive to the relevant facts in this regard (e.g., about what the evidence is, what it supports, and so on). Moreover, insofar as the referee's concern relates to relativism about truth or radical skepticism, then these are concerns that are already addressed in the paper. (It is also further argued that any educational strategy is committed to the idea of objective truth).